# Identifying and accounting for the Coriolis Effect in satellite NO$_2$ observations and emission estimates

Daniel A. Potts[1], Roger Timmis[2], Emma J. S. Ferranti[3], and Joshua D. Vande Hey[1, 4]

[1]School of Physics and Astronomy, University of Leicester, Leicester, UK
[2]Environment Agency, c/o Lancaster University, Lancaster LA1 4YQ, UK
[3]School of Engineering, University of Birmingham, Edgbaston B15 2TT, UK
[4]Centre for Environmental Health and Sustainability, University of Leicester, Leicester, UK

**Correspondence:** Daniel A. Potts (dap33@leicester.ac.uk)

**Abstract.** Recent developments in atmospheric remote sensing from satellites have made it possible to resolve daily emission plumes from industrial point sources, around the globe. Wind rotation aggregation coupled with statistical fitting is commonly used to extract emission estimates from these observations. These methods are used here to investigate how the Coriolis Effect influences the trajectory of observed emission plumes, and to assess the impact of this influence on satellite derived emission estimates. Of the sixteen industrial sites investigated, nine showed the expected curvature for the hemisphere they reside in, five showed no or negligible curvature, and two showed opposing or unusual curvature. The sites which showed conflicting curvature reside in topographically diverse regions, where strong meso-gamma scale (2 - 20 km) turbulence dominates over larger synoptic circulation patterns. For high curvature cases the assumption that the wind-rotated plume aggregate is symmetrically distributed across the downwind axis breaks down, which impairs the quality of statistical fitting procedures. Using annual NO$_x$ emissions from Matimba power station as a test case, not compensating for Coriolis curvature resulted in an underestimation of $\sim$ 9% on average for years 2018 to 2021. This study is the first formal observation of the Coriolis Effect and its influence on satellite derived emission estimates, and highlight both the variability of emission calculation methods and the need for a standardised scheme for this data to act as evidence for regulators.

## 1 Introduction

For the past three decades, national space agencies and private industry have been launching satellite-based instruments to monitor and evaluate atmospheric composition, atmospheric chemistry and anthropogenic emissions around the world. These instruments use absorption-based spectroscopy and interferometry to derive column counts of potentially harmful pollutants, and have enhanced our understanding of the impact these species have on air quality and the environment. New high-resolution instruments, such as the TROPOspheric Monitoring Instrument (TROPOMI), can resolve emission plumes from large industrial point sources such as power generation, industrial fabrication and oil refining processes (Anema, 2021; Goldberg et al., 2019; Ialongo et al., 2021; Wang et al., 2022). A suite of methods have been developed to derive emission estimates from both daily and time aggregated observations of these large sources (Beirle et al., 2011, 2019; de Foy et al., 2015; Fioletov et al., 2015; Hakkarainen et al., 2021), providing a potential avenue for these instruments to assist with regulation and to constrain bottom

up emission estimates (Marais et al., 2021; Pope et al., 2021; Potts et al., 2021). Emissions from these sources are often distinct, thermally buoyant and can extend over 0 - 20 km vertically and 10 - 200 km horizontally, where large scale atmospheric effects may progressively influence the dispersion and trajectory of the plume as it travels downwind. Here we investigate the influence of the Coriolis effect on large industrial emission plumes using observations of Nitrogen Dioxide ($NO_2$) from TROPOMI, and explore the impact of Coriolis induced curvature, plume geometry and wind fields on satellite-derived emission estimates from large point sources.

## 2 Data and Methods

### 2.1 TROPOMI $NO_2$

TROPOMI was launched by ESA in October 2017 on-board the Sentinel-5 Precursor satellite. TROPOMI is a nadir viewing (downward facing) short wave spectrometer, observing in the UV-Vis (270 – 500 nm), NIR (710 – 770 nm) and SWIR (2314 – 2382 nm) ranges (Veefkind et al., 2012). It has a spatial resolution of 5.5 × 3.5 km at nadir (7 x 5.5 km for SWIR), and a revisit time of around 13.30 local time each day. TROPOMI data products include, but are not limited to, nitrogen dioxide ($NO_2$), sulphur dioxide ($SO_2$), carbon monoxide ($CO$), methane ($CH_4$) and ozone ($O_3$), each with varying sensitivity, resolution and precision. For this study, tropospheric $NO_2$ from TROPOMI is used as it has a comparatively short photo-chemical lifetime of 2 – 24 hours (Beirle et al., 2011; Shah et al., 2020; Valin et al., 2013), so elevated tropospheric column counts are usually strongly correlated spatially with the emitting point source. This correlation enables more accurate source attribution and plume isolation for emissions from anthropogenic sources such as large industry and populous urban environments (Goldberg et al., 2020), compared to other longer lived pollutants. The TROPOMI processor upgraded to version 2.2.0 in July 2021, which resulted in a 10-15% increase in tropospheric column $NO_2$, particularly over polluted scenes with small cloud fractions (Eskes et al., 2019). Here the S5P-PAL product has been used, where observations before July 2021 have been reprocessed with the v2.2.0 processor to achieve a better retrieval and to ensure consistency across the time frame, harmonising the dataset. For this study, data from May 2018 to November 2021 was used, and observations have been over-sampled onto a 0.01 x $0.01^o$ regular grid, following the sub-pixel sampling approach of Pope et al. (2018). A quality flag of 0.75 was used as per the S5P-$NO_2$ user manual (Eskes et al., 2019), which filters out cloud contaminated pixels and poor quality retrievals. Furthermore, at least 75% of the possible pixels within a region around the source were required to pass the quality filter for that daily observation to be included in the aggregate. This region was defined as $\pm$ 20 km from the site perpendicular to the wind direction and from $-20$ km to $+60$ km along the wind direction, for each observation, which discards observations where there is not sufficient coverage over the downwind region. The number of observations included in each aggregate is annotated on the figures as "n".

### 2.2 Site selection

In order to explore the impact of the Coriolis Effect on wind rotation aggregation, several sites were investigated. Selection was based on latitude and three additional criteria: (i) 50 km from any large urban or industrial source, to avoid overlap of

Northern Hemisphere

| n | Site name | Country | Type of site | Lon | Lat | Stack height (m) | Capacity (MW) | Average surface pressure (hPa) |
|---|-----------|---------|--------------|-----|-----|------------------|---------------|--------------------------------|
| 1 | Colstrip | USA | Coal Power Station | -106.61 | 45.8835 | 215 | 1,480 | 900 |
| 2 | Janschwalde | Germany | Coal Power Station | 14.458 | 51.8344 | 300 | 3,000 | 1006 |
| 3 | Belchatow | Poland | Coal Power Station | 19.327 | 51.267 | 300 | 5,102 | 992 |
| 4 | Quassim | Saudi Arabia | Oil Power Station | 44.013 | 26.205 | n/a | 915 | 939 |
| 5 | Meh Moh | Thailand | Coal Power Station | 99.751 | 18.296 | 200 | 2,455 | 968 |
| 6 | Vinh Tan | Vietnam | Coal Power Station | 108.803 | 11.317 | 210 | 6,225 | 992 |
| 7 | Neyveli | India | Coal Power Station | 79.441 | 11.558 | 275 | 3,390 | 1002 |
| 8 | Raichur | India | Coal Power Station | 77.343 | 16.355 | 220 | 1,720 | 965 |

Southern Hemisphere

| n | Site name | Country | Type of site | Lon | Lat | Stack height (m) | Capacity (MW) | Average surface pressure (hPa) |
|---|-----------|---------|--------------|-----|-----|------------------|---------------|--------------------------------|
| 9 | Chuquicamata | Chile | Copper Smelter | -68.890 | -22.314 | n/a | n/a | 736 |
| 10 | Matimba | South Africa | Coal Power Station | 27.613 | -23.669 | 250 | 3,690 | 914 |
| 11 | Muja | Australia | Coal Power Station | 116.305 | -33.445 | 151 | 1,094 | 985 |
| 12 | Tarong | Australia | Coal Power Station | 151.915 | -26.784 | 210 | 1,400 | 962 |
| 13 | Tanjung | Indonesia | Coal Power Station | 110.745 | -6.445 | 240 | 2,640 | 996 |
| 14 | Hwange | Zimbabwe | Coal Power Station | 26.470 | -18.383 | 180 | 920 | 921 |
| 15 | Jorge Lacerda | Brazil | Coal Power Station | -48.969 | -28.452 | 200 | 857 | 1008 |
| 16 | Millmerran | Australia | Coal Power Station | 151.279 | -27.962 | 141 | 850 | 967 |

**Table 1.** Coordinates and plant information for the locations investigated in the study.

multiple plumes. (ii) Site of considerable size to produce a plume that can be detected by TROPOMI, generally > 1000 MW capacity for a power station. Note higher capacity does not equate directly to higher emissions. (iii) In operation during the 2018 – present operational lifetime of TROPOMI. Sites were identified from the Global Power Plant Database (2018) and the point source emission catalogue developed by (Beirle et al., 2021). In total sixteen sites were investigated, and their details are outlined in Table 1. Fifteen sites were coal/oil fired power stations, to allow for more direct comparisons. The remaining site is a large copper mining/smelting operation.

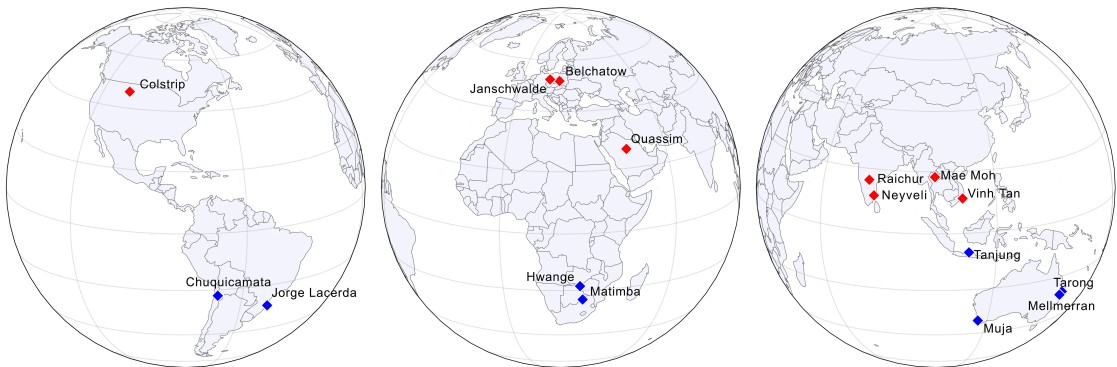

**Figure 1.** Locations of the sites used in the investigation. Northern hemisphere sites are shown in red, Southern hemisphere sites are shown in blue.

## 2.3 Wind Rotation Aggregation

Wind rotation aggregation is a well-established method for combining multiple observations whilst preserving the structure of an emission plume. Pioneered by Pommier et al. (2013), this approach has been used for various satellite based studies of point source emissions, such as cities (Goldberg et al., 2019), power stations (Fioletov et al., 2015; Hakkarainen et al., 2021), fertiliser plants (Clarisse et al., 2019; Dammers et al., 2019) and oil refineries (Potts et al., 2021). Each observation that passes quality filtering requirements (Fig. 2a) is rotated so that the wind vector is aligned to a pre-determined axis, in this case to the West-East direction (Fig. 2b). This process is repeated for every observation that passes quality filtering requirements to produce the wind rotated aggregate (Fig. 2c). The angle of rotation is found from the angle the wind vector at the origin/industrial site makes with the chosen axis of rotation. The entire observation is rotated through this angle to achieve alignment. This is done using Equation 1, where $lon_i$ and $lat_i$ are the coordinates of each pixel corner in the observation and the angle between the wind vector and the East direction is $\theta_{wind}$. This allows for all quality data to be used, and preserves the upwind-downwind profile of the emission plume (de Foy et al., 2015; Fioletov et al., 2015).

$$\frac{lon'_i}{lat'_i} = \begin{pmatrix} cos\theta_{wind} & -sin\theta_{wind} \\ sin\theta_{wind} & cos\theta_{wind} \end{pmatrix} \frac{lon_i}{lat_i} \qquad (1)$$

## 2.4 Wind data

To perform the analysis, information on the daily wind field for each observation at each site is required. As there are often no meteorological measurement stations within a reasonable distance to the sites, we therefore used modelled meteorology. Similar studies use ERA-5 reanalysis/interim wind fields, typically averaged from 0 - 500m (1000 - 950 hPa, Goldberg et al. (2019); Beirle et al. (2011)) or 0 - 800m in altitude (1000 – 900 hPa, Fioletov et al. (2015)). However, all sources investigated in this study; (a) emit from an elevated point, e.g. a 200-300 m tall chimney or stack, (b) are thermally buoyant and (c) are often

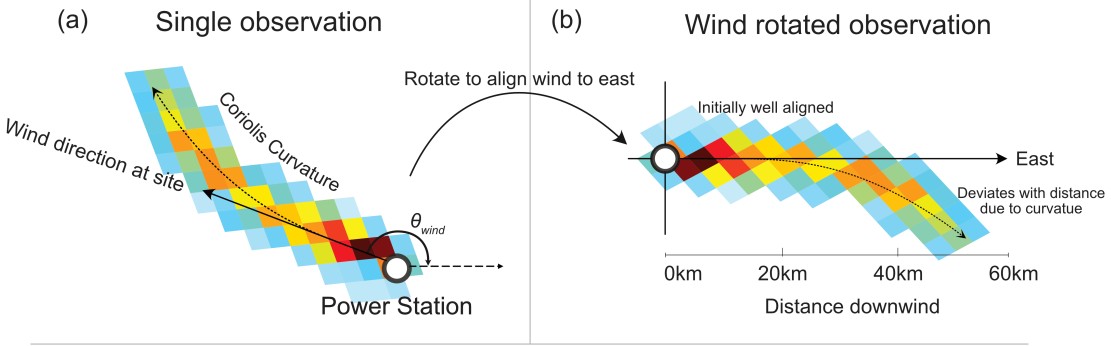

(a) Single observation

Rotate to align wind to east

Wind direction at site

Coriolis Curvature

$\theta_{wind}$

Power Station

(b) Wind rotated observation

Initially well aligned

East

Deviates with distance due to curvature

0km    20km    40km    60km

Distance downwind

(c)    Rotate and aggregate multiple observations

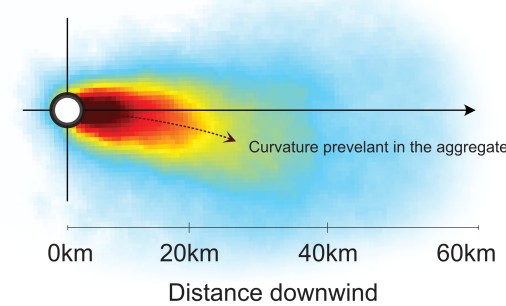

Curvature prevelant in the aggregate

0km    20km    40km    60km

Distance downwind

**Figure 2.** Illustration of the wind rotation method. (a) A single overpass from TROPOMI for Belchatow power station, on 03/06/2019, (b) plume is rotated so that its wind vector now points Eastwards. This initial stage of the plume is well aligned, but Coriolis curvature causes the latter parts of the plume to deviate from the downwind x-axis. (c) This rotational process is repeated for all quality observations and aggregated into a wind rotated average. On average the Coriolis Effect causes a clockwise deflection of the aggregate plume, increasing magnitude with distance.

in higher altitude regions with lower surface pressures than 1000hPa. For this study we find the average surface pressure at each site for the study period from the ERA-5 reanalysis, and then take the average of the wind fields from this surface pressure to a decrease of 100hPa (equating to 700-1000m altitude depending on location), in an attempt to better describe winds in the lowest kilometre of the atmosphere relative to each site. We used the ERA-5 hourly data on pressure levels (Hersbach et al., 2020), interpolated spatially using a 2D peicewise cubic approach from a $0.25°$ x $0.25°$ grid to each sites coordinates and temporally to the overpass of TROPOMI for each day. Wind speeds will vary day to day, and so the plumes included in the aggregate will each experience different ventilation/dispersion rates, and the density distribution of pollutants will vary. Furthermore, the wind speed experienced by each individual plume will vary with downwind distance and as the plume ascends vertically due to plume rise. These variations in wind speed between and within observations make it necessary to evaluate an "average" wind speed in order to infer emissions. The average needs to be both: (i) temporal (covering plumes on different days) and (ii) spatial (covering the same plume at different positions/heights within its trajectory). Only observations with wind speeds greater than

$2 \text{ ms}^{-1}$ were used to calculate emissions, as $NO_2$ decay under this condition is dominated by chemical removal rather than by wind variability, which is not the case for calmer conditions (de Foy et al., 2014).

## 2.5 Coriolis Effect

Described mathematically by Gaspard-Gustave de Coriolis in 1835, the Coriolis force is an inertial force which acts on an object moving within a rotating coordinate system. The deflection caused by this force is known as the Coriolis Effect, which manifests in the atmosphere as large-scale clockwise deflections in the Northern Hemisphere (NH), and anticlockwise deflections in the Southern Hemisphere (SH) (Figure 3). The effect is greatest at the poles and negligible at the equator, and is greater for higher velocity wind speeds.

$$F_x = 2 \cdot \Omega \cdot sin(\phi) \cdot v$$
$$F_y = 2 \cdot \Omega \cdot sin(\phi) \cdot u$$
(2)

Where F is the horizontal components of the Coriolis force, $\Omega$ is the Earths rotation, $\phi$ is latitude and (v, u) are the horizontal components of the wind. $NO_2$ emissions from large industrial sources can extend tens of kilometres, and can rise quickly due to their thermal buoyancy. Emission plumes do not move independently or freely within the atmosphere, but instead follow the given wind field. This wind field is influenced in part by the Coriolis Effect, and so it follows that observed emission plumes of significant magnitude will be deflected due to the Coriolis Effect as they move with the air mass. Furthermore, as the plume ascends due to its thermal buoyancy and the wind fields inherent vertical velocity, the plume will move to a degree with an Ekman Spiral (Ekman, 1905; Åkerblom, 1908), which itself is a consequence of the Coriolis Effect and is demonstrated

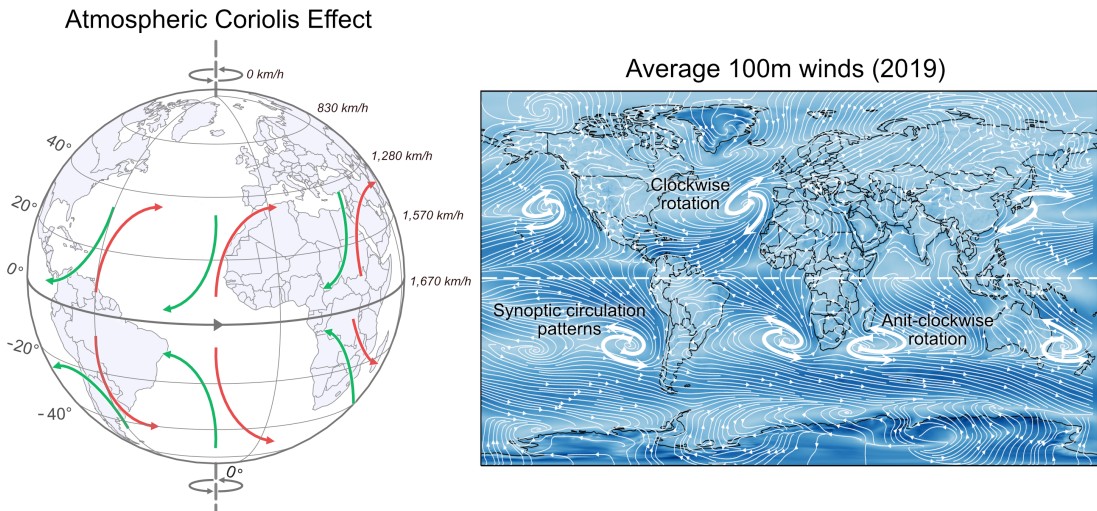

**Figure 3.** Illustration of the Coriolis Effect on atmospheric circulation patterns. The right hand plot is produced using an average of the ERA-5 100m winds for 2019 at 12pm

## Atmospheric Ekman Spiral

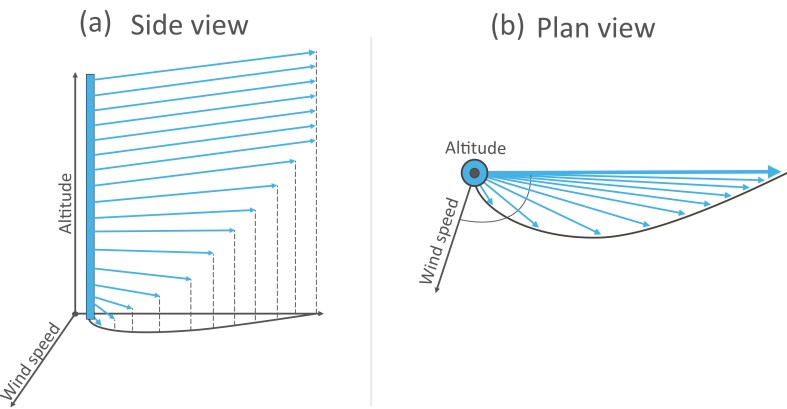

**Figure 4.** Schematic describing the process behind an Ekman spiral from (a) side and (b) plan views. This diagram shows the spiral for the Southern hemisphere.

in Figure 4. The boundary layer can be considered in three parts. The lowest "laminar sub-layer" only concerns the very near surface, above which is the Prandtl layer (20 - 100m) where surface turbulence is fully developed but without influence from Earths rotation, and then the Ekman layer (100 - 1000 m) where movement is driven by the balance between pressure gradients, frictional effects and the Coriolis force (Holton, 2004; Marshall and Plumb, 2016), and is strongly dependant on thermal stability (Sorbjan, 2003). Surface frictional forces bring the wind speed towards zero at the surface, and as frictional effects decrease with height and along the pressure gradient, wind speed increases towards the geostrophic wind. As the Coriolis force is a function of wind speed, shown in Equation 2, and as wind speed increases with altitude, winds at the surface are orientated at an angle to winds further up the vertical column until laminar flow is reached near the boundary layer, creating the Ekman spiral. Whilst a mathematically pure Ekman spiral is rarely observed in a real atmosphere, the Ekman layer does impart a degree of curvature within the boundary layer relative to the geostrophic wind, increasing in angle with depth (Holton, 2004). The plumes in question are thermally buoyant, and inserted from a 100-300 m chimney into the Ekman layer, where their curvature will be modified both by the Coriolis force horizontally, but also via the Ekman spiral as it travels vertically with the air mass. This introduces variability in the magnitude of deflection, as magnitude is dependent on altitude, and the altitude the plume reaches is dependent on its exit velocity, temperature and most importantly the meteorology, which will vary day to day. Plume curvature becomes an issue when observations are rotated into an upwind-downwind aggregate to derive emissions, as the averaged plume may exhibit strong curvature and be unevenly distributed to one side of the common downwind axis. This deflection can be seen in some previous wind rotated averages in other works, such as Figure 4 of Hakkarainen et al. (2021), though its presence is not discussed. Examples of strong emission plume curvature from two of our investigated sites are given in Figure 5. Whilst this study focuses on aggregate observations and the implications of curvature on emission estimates, further investigation into the conditions that produce extreme curvature in daily observations is required. High curvature cases

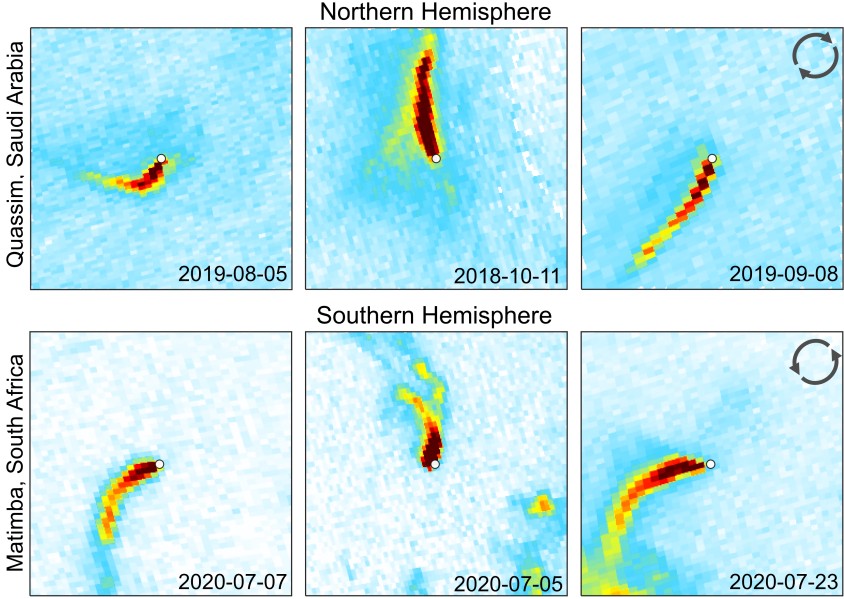

**Figure 5.** Example of Coriolis influence on NO$_2$ emission plumes from daily TROPOMI NO$_2$ observations above Quassim, Saudi Arabia and Matimba, South Africa. Power stations are shown as the central white dot of each observation.

like these are likely a combination of high wind speed as well as favourable conditions for plume rise, where the Coriolis force and the Ekman spiral both contribute to the observed curvature. Meteorological dispersion modelling would be required to explore this, which is beyond the scope or aims of this study. Effects such as Coriolis curvature and Coriolis induced
Ekman spirals are just two of the many influences on the movement of the atmosphere, and various micro/meso/synoptic scale processes all contribute with variable intensity. Consequently, wind fields are not always dominated by the Coriolis effect, and can circulate in the opposing direction to the Coriolis force, or not at all, and so not every observation will show the same magnitude or "expected" direction of curvature for that hemisphere. However, on average we expect emission plumes to curve in favour of the direction of the Coriolis force for that hemisphere.

**2.6   Curvature fitting**

In order to evaluate the curvature of the wind rotated plume, a "spine" was fitted to the aggregate. Firstly, the across-wind line density at 1 km transects perpendicular to the downwind axis were taken, as shown in Figure 6a. These profiles show a characteristic normal distribution, with the maxima transect located near the origin. The origin here corresponds to the west-east "downwind" axis used for the wind rotated aggregate. At greater distances downwind, the maximum of each transect laterally
deviates with increasing distance from the origin, due to the curvature introduced by the Coriolis Effect. These profiles contain a degree of noise, and so a Gaussian smoothing procedure is applied, displayed in Figure 5b. This allows for the maxima

of each transect to be more readily identified above the per-pixel variability. These peaks are then fitted with a second order polynomial to identify the "spine" of the plume and the degree of deviation from the origin, shown in Figure 6b as a dashed red line. This "spine" is shown in later figures as a dashed black line.

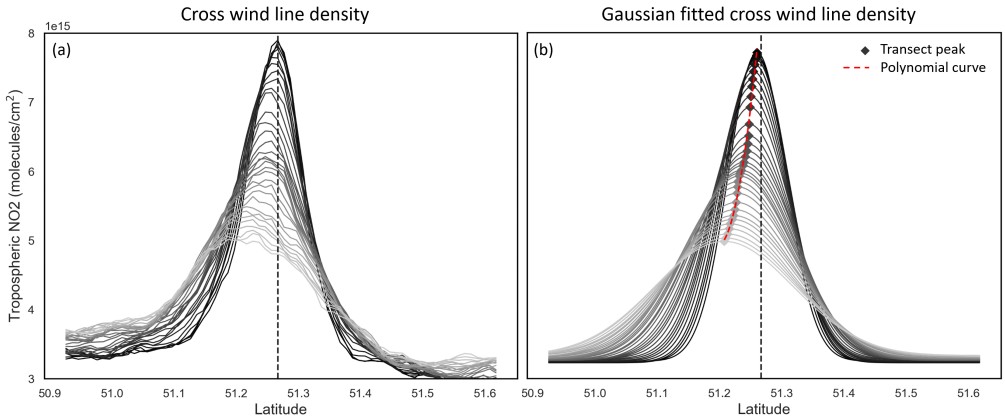

**Figure 6.** Across wind line density profiles (a) and Gaussian fitted line density profiles (b) taken at regular 1 km transects for the wind rotated average of Belchatow Power station in Poland. The transect at the origin is given by the black line, with the gradient getting lighter as distance increases from the origin. Produced using data from May 2018 – November 2021.

## 2.7 Emission estimation via an Exponentially Modified Gaussian (EMG)

To derive emissions from a wind rotated aggregate, an Exponentially Modified Gaussian (EMG) (Beirle et al., 2011; de Foy et al., 2015; Fioletov et al., 2015; Goldberg et al., 2019) is fitted to the integral of the across-wind line densities in Figure 6, the form of which is given in Equation 3.

$$NO_2 \ Line \ Density = \alpha \left[ \frac{1}{x_o} exp \left( \frac{\mu}{x_o} + \frac{\sigma^2}{2x_o^2} - \frac{x}{x_o} \right) \Phi \left( \frac{x - \mu}{\sigma} - \frac{\sigma}{x_o} \right) \right] + \beta \tag{3}$$

$$NO_x \ Emissions = 1.33 \left( \frac{\alpha}{\tau_{eff}} \right), \quad where \ \tau_{eff} = \frac{x_o}{\omega} \tag{4}$$

Where $\alpha$ is the total number of $NO_2$ molecules minus the background, $\beta$. $x_o$ is the e-folding distance downwind from the source, $\mu$ is the displacement of the apparent source relative to the assumed source center, $\sigma$ is the standard deviation of the Gaussian function and $\Phi$ is the cumulative distribution function. Using a non-linear iterative least squares fitting approach, $\alpha$, $x_o$, $\sigma$, $\mu$, and $\beta$ are determined. This process is illustrated in Figure 7. From these fitted parameters we can then calculate an effective lifetime, $\tau_{eff}$, using the mean wind speed, $w$. Equation 4 is then used to calculate $NO_x$ from the TROPOMI derived

NO$_2$, using $\tau_{eff}$ and a scaling factor of 1.33 to convert from NO$_2$ to NO$_x$. This scaling factor describes the typical NO$_2$/NO ratio under polluted conditions at noon (Seinfeld and Pandis, 2016).

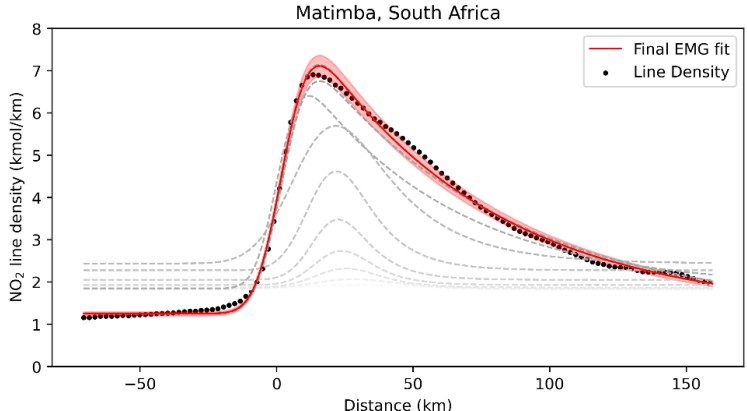

**Figure 7.** Demonstration of the Curvefit process of fitting the Exponentially Modified Gaussian (EMG) to the NO$_2$ line density. Produced using wind rotated NO$_2$ from Matimba power station for May 2018 – Nov 2021. The red line shows the final EMG fit, and the grey lines illustrate the iterative fitting procedure, changing from light grey to black as it converges to the final EMG fit.

## 3 Results and Discussion

### 3.1 Wind rotation aggregates of selected sites

Figure 8 shows both the unrotated and wind rotated aggregates for Belchatow power station in Poland (a-b) and Matimba power station in South Africa (c-d), as examples of a Northern and a Southern site. The black dashed line shows the "spine" of the plume and the value of "n" shows the number of observations included in the aggregate. Both sites are strong emitters, well isolated from other sources and the surrounding area consists of relatively simple topography. Both of these sites show strong curvature in their wind rotated aggregates, and whilst the plume's structure is successfully preserved, there is a clear distinguishable curvature in favour of the Coriolis force direction. As these two example sites are non coastal and in regions with simple topography, there are minimal micro/meso scale processes influencing the wind field, and so the effect of the Coriolis force is easily identified, as it more frequently prevails over other influences. The remaining sites are shown the appendix (Figures A1-A3), grouped by their hemisphere. Of the sixteen sites investigated, nine showed the expected curvature for the hemisphere they reside in, varying in magnitude. Five showed no or negligible curvature, and two showed opposing or unusual curvature. These latter two sites are located in areas with steep and highly variable topography which may "steer" plumes locally, and such "steering" may dominate over larger-scale Coriolis curvature. A good example of this is given in Figure 9. The Jorge Lacerda power station is located in Brazil at a latitude of -28.45 in the Southern hemisphere, yet we observe clockwise, Northern hemisphere curvature. This can be explained by the topographical surroundings of Jorge Lacerda, as the

175 power station sits between the South Atlantic and the Serra do Mar coastal mountain range, where there is an abrupt +1200m increase in altitude over a short distance. Onshore synoptic-scale winds and sea breeze penetrate inland and are "steered" by the topography. These local effects, over tens of kilometers, dominate over larger synoptic weather patterns, and therefore outweigh the influence of the Coriolis Effect, leading to this unexpected display of Northern hemisphere curvature.

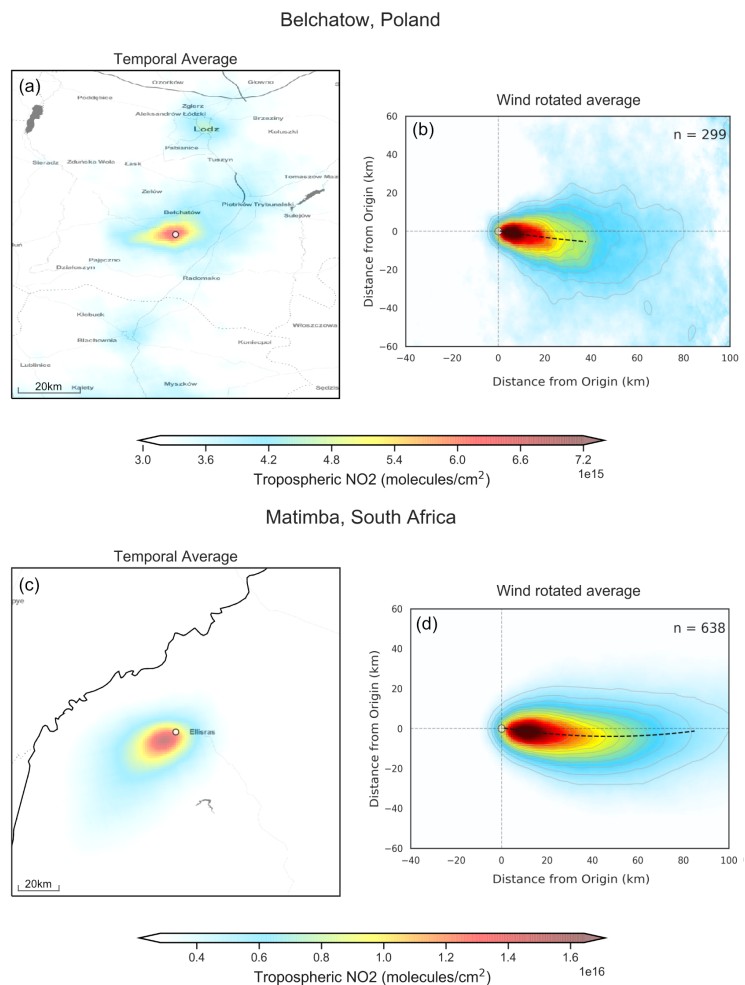

**Figure 8.** Unrotated and wind rotation averaged tropospheric $NO_2$ columns from TROPOMI, using data from May 2018 to December 2021 for (a-b) Belchatow power station in Poland and (c-d) Matimba power station in South Africa. Map tiles by Stamen Design, under CC BY 3.0. Street data by OpenStreetMap, under ODbL.

## 3.2 Impact of selected wind level on quality of aggregate

To investigate the influence of the chosen wind field on the final aggregate and emission value, emissions were derived for Matimba using wind fields corresponding to pressure levels: 900 hPa (~100m), 875 hPa (~250m), 850 hPa (~450m), 825

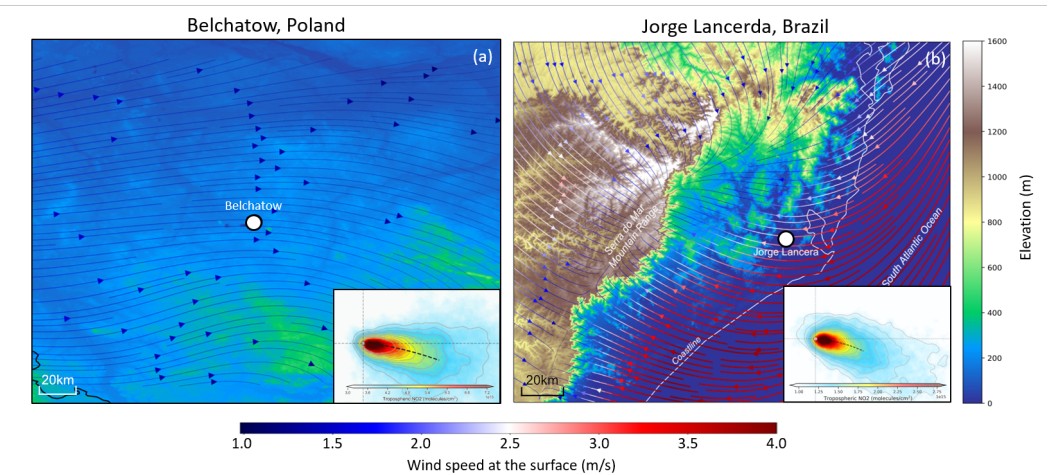

**Figure 9.** (a) Average 2018 surface wind fields from ERA-5 Reanalysis, including topography from the Copernicus GLO-30 digital elevation model for (a) Belchatow in Poland and (b) Jorge Lacerda in Brazil. Included in the bottom right of (a) and (b) is the wind rotated tropospheric NO$_2$ columns from TROPOMI, using data from May 2018 to December 2021.

hPa ($\sim$700m) and 800 hPa ($\sim$ 1000m), as well as an average of all six levels. As seen in Figure 10, with increasing altitude the plume migrates anticlockwise, e.g. the plume spine is below the aggregate x-axis at 900 hPa ($\sim$100 m) but above it at 800 hPa ($\sim$1000m). Due to the Ekman spiral, winds near the surface are orientated clockwise (for the Southern Hemisphere, as used in this example) from winds at higher altitude, and so rotational alignment using winds at lower altitude results in a clockwise deviation compared to alignment using winds at higher altitude, which is to be expected. The initial stages (first 10km downwind) of each plume align with the aggregated x-axis and are laterally symmetric around it. This confirms that rotations based on the wind vector near the source produce well-aligned and symmetric aggregate plumes in the near field. However, at greater distances the Coriolis curvature becomes distinguishable from the initial alignment, and as the plume progresses down-wind it is increasingly deviated and asymmetric relative to the common axis. Aggregates using winds at 900hPa – 850hPa are very similar, with 825-800hPa showing better initial alignment with the axis of aggregation, within the first 10km. This initial agreement may be due to the wind direction at these levels being the most representative, or due to an inherent bias in the ERA-5 model, as there may be small but consistent differences between the wind at the site compared to the coarse $0.25^o \times 0.25^o$ modelled fields. All exhibit comparably strong curvature, which shows that curvature is not an artefact of the wind product but an inherent feature of the observations. Furthermore, wind speed increases with altitude, and as wind speed factors into the emission calculation it is important that representative wind fields are used, rather than selecting purely by the field that yields the best initial alignment. For the purposes of this study the average wind field is used, as in most cases it results in good plume alignment whilst also reasonably describing the wind speeds experienced as the plume travels downwind both vertically and horizontally from the source. Although the 800hPa aggregate has the best initial alignment, these higher wind speeds would not be experienced by the plume for the majority of its lifetime, and would lead to an overestimate in the

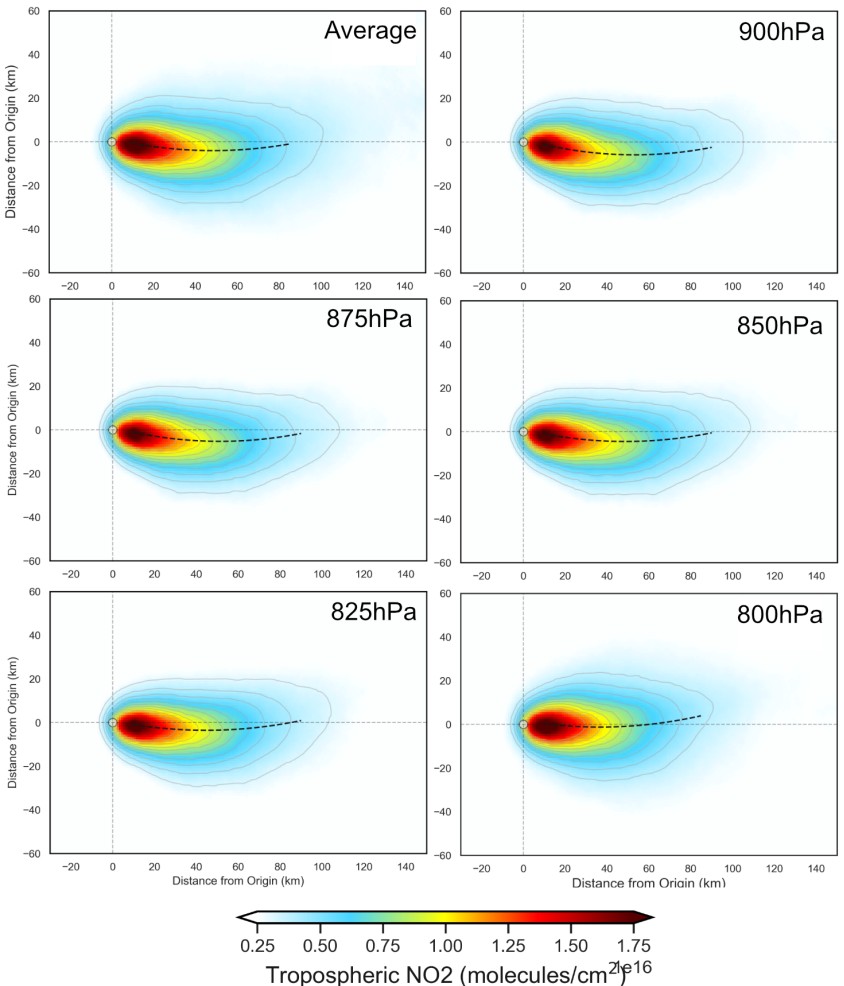

**Figure 10.** Demonstration of the difference in aggregate when using wind products from different pressure levels, using data from Matimba power station for May 2018-November 2021

emission, as demonstrated by Figure A4. There must be a trade off between the geometric alignment of the aggregated plumes and ensuring the wind field reasonably describes the wind speeds experienced.

### 3.3    Impact of Coriolis curvature on emission estimates

From the wind rotated aggregate, the typical next step is to take the integral of evenly spaced (1 km) across-wind ($\pm$ 30 km) segments perpendicular to the x-axis, as shown in Figure 11(a). This approach assumes the wind rotated plume is distributed evenly either side of the common axis. Occasionally this assumption holds, as the curvature is often minor/negligible and so

emission estimates are marginally impacted. However, as evident with sources such as Matimba, this is not always the case, and the plume can deviate considerably from the x-axis due to the plume's inherent curvature. Using the EMG emission estimation method discussed in Section 2.7, we calculated $NO_x$ emissions for Matimba under two scenarios; (a) using cross wind segments perpendicular to the common downwind axis, and (b) using cross wind segments perpendicular to the curved spine of the plume. Scenario (b) aims to counteract the influence of the plume's geometry on the emission estimate, by re-centring the integral along the curved spine of the plume. Uncertainties are determined using a bootstrapping approach, whereby observations are randomly selected, with replacement, to be included in the aggregate. Each scene has both measurement and numerical error, and so by assuming these errors are randomly distributed, the random selection and replacement of scenes in the bootstrapping algorithm allows for the estimation of the impact that all these errors have on the emission estimate (de Foy et al., 2015, 2014).

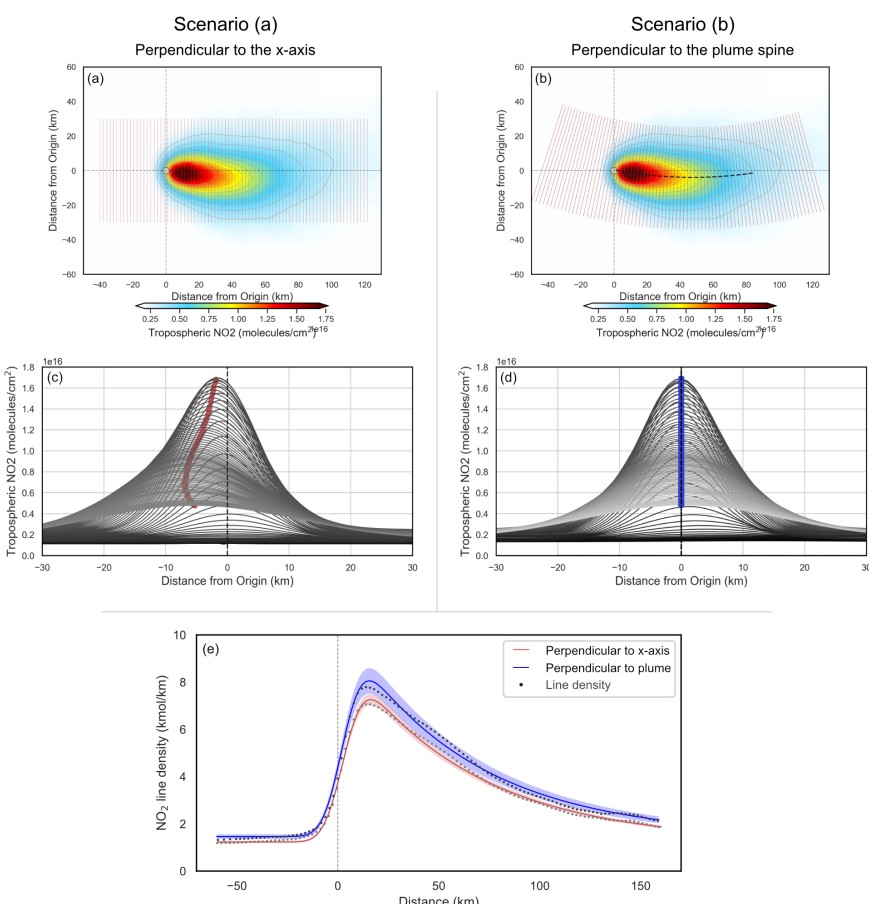

**Figure 11.** Demonstration of the impact the Coriolis Effect has on the resulting emission estimate. (a) & (c) show the results using cross sections perpendicular to the x-axis, whereas (b) & (d) show cross sections perpendicular to the plume of the spine. (e) Shows the EMG fit for each scenario with the shaded region showing the quality of each fit.

Uncertainty introduced by the selection of wind field is evaluated through sensitivity tests, where emissions are calculated using each pressure level, shown in Figure A4. This approach does not account for uncertainty introduced due to the clear sky bias, caused by only using cloud-free observations. Figure 12 shows annual $NO_x$ emission estimates for Matimba power station from; (i) a similar study of Matimba using TROPOMI $NO_2$ and EMG before S5P-PAL released (Hakkarainen et al., 2021), (ii) scenario (a), (iii) scenario (b) and finally (iv) reported values from the site operator, Eskom, derived from Continuous Emission Monitoring Systems (CEMS) (https://www.eskom.co.za/dataportal /emissions/ael/matimba-c2/). The reported emissions are not provided with an uncertainty, and so conclusive statements about the accuracy of the TROPOMI based estimate are not possible.

The use of S5P-PAL explains the increase in emissions between Hakkarainen et al. (2021) and scenario (a), as S5P-PAL can lead to a 10-15% increase in tropospheric columns for polluted cloud free scenes (Eskes et al., 2019). This translated to a 5-10% increase in the emission estimate for Matimba power station. Between scenarios (a) and (b) there is a substantive $9.1\%$ $\pm 1.92\%$ increase in emissions annually on average, for the years 2018-2021, when the curved geometry of the wind rotated plume is taken into account. Scenario (b) yields an emission value closer to the reported value and its uncertainty is within range of the reported emissions for 3 out of the four years investigated. This strongly suggests an improved satellite derived emission estimate can be achieved by considering the curvature of the wind rotated aggregate. This approach is not constrained to the Coriolis effect, as any consistent misalignment of the aggregated plume, such as the topographic deviation seen at Jorge Lacerda, could be accounted for when calculating emissions by following the method discussed here. This approach is rather generalised, and could be easily applied autonomously to a given source. We suggest that the spine fitting and along-

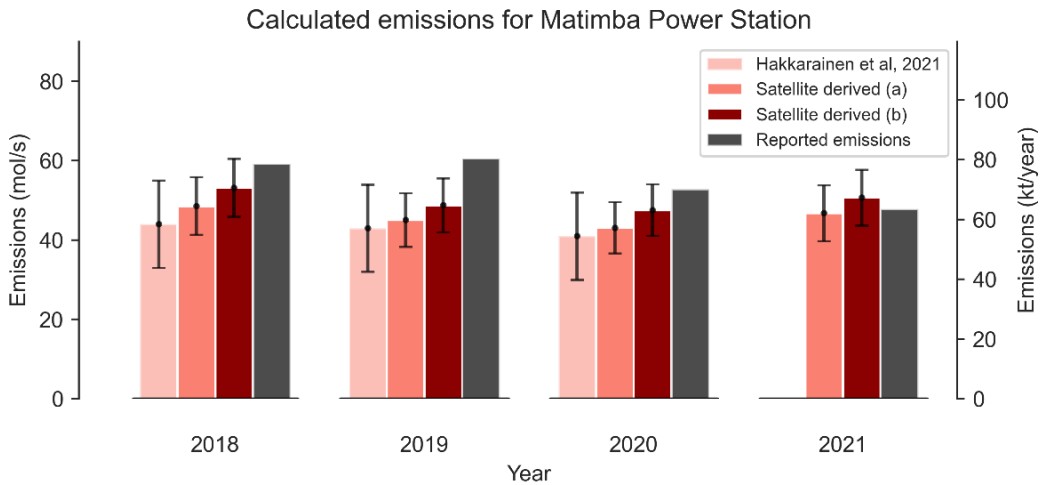

**Figure 12.** Comparison of emission estimates from TROPOMI $NO_x$ using different applications of with the EMG method, compared to values obtained by Hakkarainen et al. (2021) for the same site, and emissions reported from estimates by the operator, Eskom (https://www.eskom.co.za/dataportal/emissions/ael/matimba-c2/).

spine integration steps should be incorporated into a regulatory wind rotation aggregation approach, in order to minimise the
influence of plume curvature on the emission estimate.

## 4   Conclusions

This study demonstrates the Coriolis Effect's varying influence over the trajectory of point source emission plumes observed
by TROPOMI, and has shown how strong curvature can lead to substantive underestimations in the emission estimate if it
is not accounted for. Of the 16 locations investigated, nine showed the expected curvature for the hemisphere they reside in,
varying in magnitude. Five showed no or negligible curvature, and two showed opposing or unusual curvature. The sites which
showed conflicting curvature are all within regions with complex terrain where air flows are steered by local topography in
ways that dominate over larger-scale influences such as the Coriolis Effect. Emissions of $NO_x$ were estimated for Matimba
power station in South Africa, chosen as a demonstration due to its strong curvature and good data coverage. Conducting the
emission calculation in a way that accounted for the inherent curvature of the plume resulted in an average $\sim 9\%$ increase in
yearly emitted $NO_x$ over the regular approach, and was more comparable to and within the uncertainty range of the emission
value reported by the operator. As demonstrated, the wind rotated aggregate of a source is not always aligned and distributed
along the common downwind axis, and so site specific considerations need to be included. This study formally identifies, for
the first time, Coriolis curvature in the satellite record, and suggests how it can be accounted for during emission analysis of
high curvature cases, such as Matimba power station. Considerable work remains in understanding the conditions that drive
the more extreme curvature cases, such as those shown in Figure 5. The dynamics of the atmosphere are well researched, but
the behaviour of these variable emission plumes released from height at considerable temperatures, is less so, especially for
characterising their trajectory and curvature. Greater understanding is required about the combined and variable contributions
of horizontal Coriolis curvature with the additional curvature induced by the Ekman spiral, which is essential for correctly
interpreting observations of these plumes in the satellite record. For satellite evidence to be used by regulators and operators,
there needs to be a standardised data processing routine in place for emission calculation and uncertainty analysis, as there is
with air quality modelling, so that satellite observations can be used to generate consistent and auditable evidence of emissions
for regulatory purposes. The rapid development of satellite instruments over the next decade offers a unique opportunity for air
quality regulators and industrial operators to begin to monitor emission performance remotely and persistently, and so a greater
understanding of the role atmospheric dynamics has on satellite derived emission estimates is vital.

*Data availability.*  The TROPOMI data used in the findings of this study are freely available at the following: https://s5phub.copernicus.eu/
dhus/#/home (accessed on 1 February 2022). Site selection information was obtained from the Global Power Plant Database publicly available
at https://datasets.wri.org/dataset/globalpowerplantdatabase (accessed 1 July 2022) and from the public database produced in Beirle et al.
(2021). ERA5 Reanalysis products (Hersbach et al., 2020) was downloaded from the Copernicus Climate Change Service (C3S) Climate
Data Store, and are publicly available at https://cds.climate.copernicus.eu/cdsapp#!/dataset/reanalysis-era5-pressure-levels (accessed on 1

July 2022). Emissions data for Matimba power station were obtained from the operator's website, available at https://www.eskom.co.za/dataportal/emissions/ael/matimba-c2/ (accessed on 1 July 2022).

*Author contributions.* Conceptualisation: D.A.P., R.T., E.J.S.F. and J.D.V.H; methodology: D.A.P., R.T. and J. D. V. H.; investigation: D.A.P.; formal Analysis: D.A.P.; writing—original draft preparation: D.A.P.; writing—review and editing: D.A.P., E.J.S.F., R.T. and J.D.V.H.; supervision: E.J.S.F., J.D.V.H. and R.T. All authors have read and agreed to the published version of the manuscript.

*Competing interests.* The authors declare no conflict of interest.

*Disclaimer.* The views expressed are those of the authors, and are not formal positions of their organisations.

*Acknowledgements.* This research and D.A.P. is supported by the CENTA Doctoral Training Partnership (UK Natural Environment Research Council, NERC) (NE/S007350/1), in CASE partnership with the Environment Agency. J.D.V.H. acknowledges funding from the NIHR HPRU in Environmental Exposures and Health at the University of Leicester. E.J.S.F. acknowledges funding from the NERC Knowledge
Exchange Fellowship MEDIATE (NE/N005325/1). The University of Leicester High Performance Computing Facility ALICE was used to conduct data processing and analysis. The authors acknowledge the TROPOMI mission scientists and associated Sentinel-5P personnel for the production and distribution of the TROPOMI data products. The analysis contains modified Copernicus Climate Change Service information 2020. Neither the European Commission nor ECMWF is responsible for any use that may be made of the Copernicus information or data it contains.

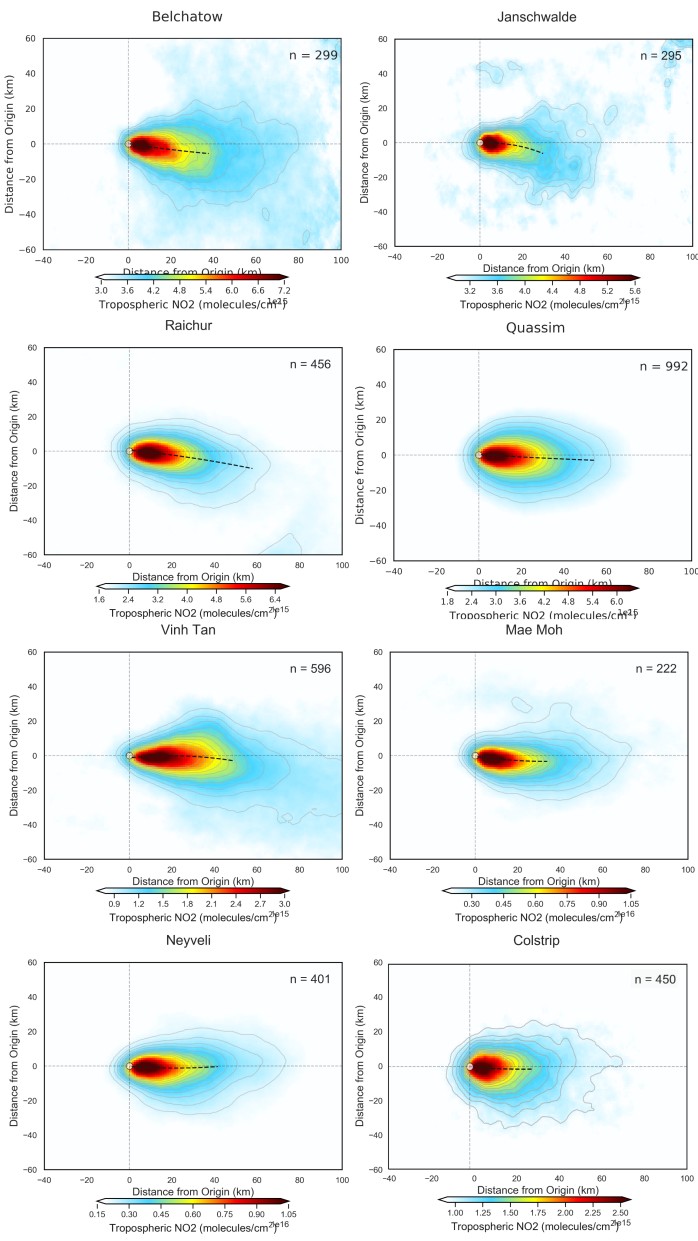

**Figure A1.** Wind rotated aggregates of all Northern hemisphere sites, with the plume spine signified by the black dashed line.

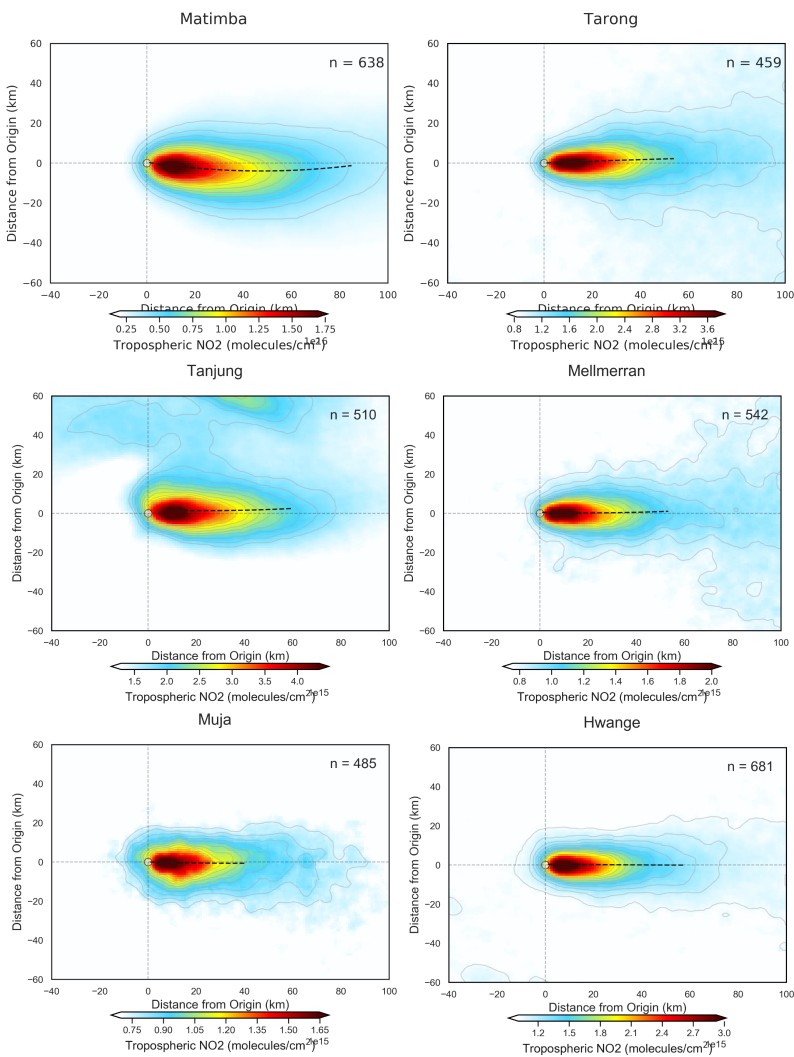

**Figure A2.** Wind rotated aggregates of all Southern hemisphere sites, with the plume spine signified by the black dashed line.

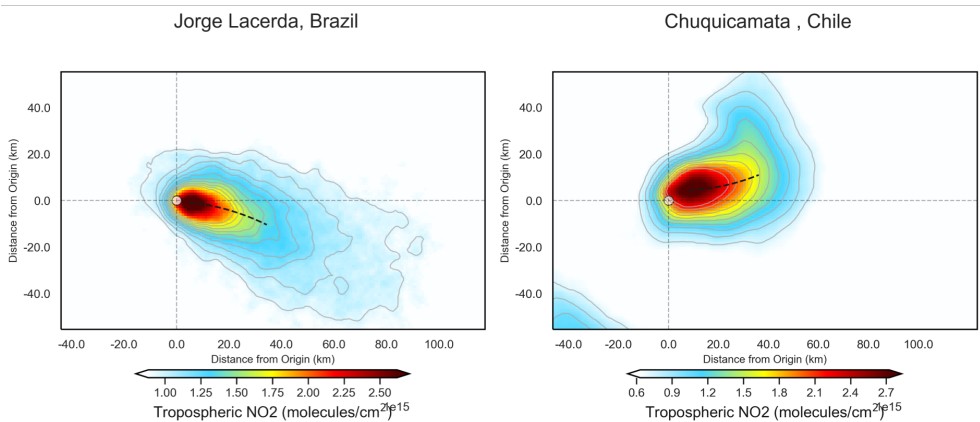

**Figure A3.** Wind rotated aggregates of the non-conforming sites, with the plume spine signified by the black dashed line.

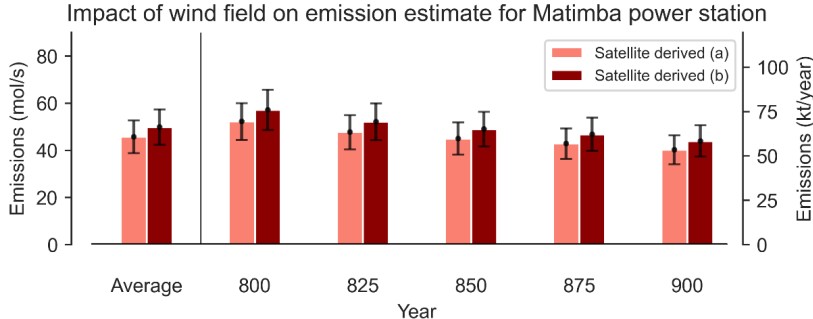

**Figure A4.** Demonstration of the influence the chosen wind product has on the final emission estimate using data for the entire 2018-2021 period from Matimba power station.

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
