# Peer review of "Identifying and accounting for the Coriolis Effect in satellite NO2 observations and emission estimates"

_Atmospheric Chemistry and Physics, 2022_

## Author Response (AR1)

**Reviewer 1**

We would like to thank the reviewer for their valuable comments and constructive feedback on our manuscript. Their suggestions have greatly improved the quality of the paper and we appreciate the time and effort they took to review our work. Thank you.

Reviewer comments have been copied (R:), and answered (A:).

R: In the abstract, results and conclusions you mention studying 17 sites, 9 of which show an effect, 5 do not, and two are unusual. This adds up to 16 sites – what happened to the last one?

A: The site at Aksu (included in the table) was omitted mid-analysis as we found to be near another major source, which was unanticipated, leading to a very contaminated wind rotated plume. We therefore removed it from the analysis, but forgot to remove it from the table, and are grateful that the reviewer spotted this. Figure 1 has also been updated to reflect this. Through checking this, we discovered that Melmerran was included twice in the supplementary material, and Colstrip was accidentally omitted, and so this was rectified.

R: Section 2.1 – More detail is needed on what data you used from the TROPOMI record. What time period are you using? Is it all available data (which isn't straight from launched in October 2017, but spring 2018), or a subset? Also, what sort of region to use to cover these plumes? Does this change depending on the site choice? Roughly how many observations do you get per site? (If you include this last point, I would perhaps calculate a rough statistic based on the percentage of cloudy days in a sample of days if the full dataset is difficult to process)

A: Information regarding the date range, the definition of a "region" around the site and information on the number of observations used in the aggregates has been added to the manuscript in section 2.1, in and around lines 44-50.

R: Line 57 – Are the 17 sites chosen the only ones available that match all the criteria or are these a sub section?

A: The sites selected are a subsection of the available sites. Whilst there was a surplus of northern hemisphere sites that match the criteria, the number of southern hemisphere sites that matched the criteria was limited, and so in order to maintain an even split between hemispheres we chose a subsection of available sites.

R:Figure 2 – this looks like it could be useful to the reader but not referenced in the text anywhere

A: Figure 2 has now been referenced and explained in section 2.3

R: A general comment is that it would be good to expand on how this can be used more widely (possibly in the conclusion). You mention in the paper how your calculations could be used by regulators and operators but what steps are needed between your case studies and a more general approach? Could your method be applied to plumes across the globe and not require manually checking each one?

A: Lines 207-214 were added to address this valuable comment. We hope this is sufficient.

R: Related to this, how do you determine what counts as 'expected' when looking at the plume? Is this the author's judgement or is there a quantitative statistic?

A: "Expected" curvature is determined by the hemisphere the site resides in, as we would expect that if Coriolis does impart a degree of curvature then it would follow the direction of that hemisphere.

R: Some discussion in section 4 on the impact of time-period of the plume aggregation would be good to see. I assume you've used all possible plumes, but do you think this technique would work with a

years' worth of data? Or a month's? (Assuming good data). Could this even be used for any curving plume from a single day?

A: Yes we have used all possible plumes that pass the quality filtering requirements during the time period for the aggregates shown in the supplementary information. The annual emission values were produced using yearly aggregates for 2018-2021, and so the method works well for yearly timeframes.

R: General formatting – There are a few occasions where the references aren't in chronological order

A: The referencing style was taken from the ACP Latex template, which uses an alphabetical bibliography. We checked the chronology of multiple papers with the same author and could not find an inconsistency. Happy to hear any suggestions if there is a problem we have missed.

R: General formatting – There are inconsistencies with the $NO_2$ subscript (e.g. figure 5, figure 6 caption, figure 9, figure 10, line 117) which need addressing.
A: This has been addressed.

**Reviewer 2**

We would like to thank the reviewer for their insightful comments and constructive feedback on our manuscript. Their suggestions have greatly improved the quality of the paper and we appreciate the time and effort they took to review our work. Thank you.

Reviewer comments have been copied (R:), and answered (A:).

R: The movement of a power plant plume obviously cannot be considered as an active and free movement in an initial direction that just changes due to Coriolis force: The plume follows the given wind fields, and these are not generally always/everywhere curved, as noted by the authors and also illustrated in Fig. 3. This aspect should be pointed out clearly.

A: This point is gratefully received and is addressed in lines 98-100. We hope this is a sufficient and clear explanation of the process.

R: There is a quite striking difference between the showcase examples in Fig. 4, where plume direction changes significantly within rather short distance, and the mean rotated plume in Fig. 7, where the overall effect is rather weak. I.e. besides the examples in Fig. 4, there are many days without observable plume curvature or even opposite direction. It would be important to understand why the effect is strong on some days and not present on others, and I would like to ask the authors to look for key variables that might explain this different behaviour (e.g., season, pressure & temperature). And, as the Coriolis force itself is well understood, it would be desirable to relate the observed curvature to the actual Coriolis force. If this is not possible, please discuss why.

A: We are grateful for this comment, as we discussed this internally during the planning phase of the study. We decided that the current paper would focus on the impact of Coriolis curvature on the quality of the emission estimate, rather than a statistical analysis of which conditions are conducive to strong Coriolis curvature on an individual plume. The above would be a valuable and interesting study, but would require a considerable amount of additional analysis and deviation from the intended scope of the current paper. Therefore, we are strongly considering following this in a separate study, more focussed on meteorology and atmospheric dynamics rather than emission estimates and regulatory implications. We hope this is satisfactory for the reviewer.

R: I am also missing a discussion of the Ekman spiral:
https://glossary.ametsoc.org/wiki/Ekman_spiral
https://www.researchgate.net/publication/228751584_Air_pollution_meteorology
Perhaps the convincing examples rather show the Ekman spiral: during plume rise, wind speed

increases, and from simple assumption of geostrophic winds, this implies a spiral (caused by Coriolis force).

A: Acknowledgement and discussion of the Ekman spiral has been added to section 2.4, lines 100-106, and we are very grateful to the reviewer for identifying this gap in the explanation/discussion of the driving processes in plume curvature.

R: Minor comment: Fig. 4: Please add a km scale or provide lat/lon on the axis.

A: Km scale has been added to all maps

---

## Author Response (AR2)

**Reviewer comments:**

**R:** The authors revised the manuscript according to the reviewers' comments. However, some modifications were made with quite minimalistic effort. Concerning a central issue raised in my original review:

*"There is a quite striking difference between the showcase examples in Fig. 4, where plume direction changes significantly within rather short distance, and the mean rotated plume in Fig. 7, where the overall effect is rather weak. I.e. besides the examples in Fig. 4, there are many days without observable plume curvature or even opposite direction. It would be important to understand why the effect is strong on some days and not present on others, and I would like to ask the authors to look for key variables that might explain this different behaviour (e.g., season, pressure & temperature). And as the Coriolis force itself is well understood, it would be desirable to relate the observed curvature to the actual Coriolis force. If this is not possible, please discuss why".*

The authors refer to possible future studies. I do not expect the current paper to answer all open questions; however, I definitely expect the authors to extend the discussion and at least mention these open questions and current shortcomings of our understanding. The discrepancy between the extreme showcases and the average curvature has to be pointed out. Concerning the Ekman spiral, this has just been added sloppily. It is now mentioned in lines 100 ff, but is not discussed any further in the rest of the paper. No references are provided. Do the authors think that the showcases they found are showing an Ekman spiral? This should at least be mentioned in the discussion and/or conclusion.

**A:** We apologise for the shortcomings of our reply to this comment and we fully accept the reviewers concerns with how we addressed the original comment. We are grateful to the reviewer for re-iterating this point, as the improvements made have re-enforced our understanding and improved our discussion of the processes involved. We have supplemented the following sections, highlighted in blue, including references.

- Section 2.5
- Section 3.2
- Section 4

**R:** A new figure was added as part of the existing Fig. 4. This suggests that Fig. 4c actually shows Ekman spirals - is this the intention of the authors? If not, I recommend to separate the figure of the Ekman spiral. In the current version, the figure caption has not been modified and does not describe panels (a) and (b).

- The figures have been separated and issues with the caption have been fixed

**Editor comments:**

**R:** Clarify why we need to track to the plume to the distances where the Coriolis Effect becomes evident to determine the emissions.

**A:** In order to correctly quantify emissions from an aggregate of a plume, we much consider the plume along its entire trajectory in order for the chemical lifetime assumptions and Gaussian fitting procedure to work. Omitting a section of the plume would be discarding results and would lead to an underestimation of emissions. It just so happens that the plumes lifetime allows for transport over a range where the Coriolis force begins to act on it. We do not consider the whole plume so that the Coriolis force can be seen, but rather we must consider the whole plume to properly quantify emissions, and across this range we get curvature due to the Coriolis force. Another reason for tracking/evaluating plumes over the whole length (10skm) is that these plumes show emissions averaged over several hours of discharge and dispersion.  The resulting emission estimates are therefore more likely to be a representative sample of the source performance, compared to a short section of the plume (say, <1km) which is more of a "snapshot" that may not be so representative. Effective regulation requires representative evidence, so that emission estimates based on longer plumes are more compelling as evidence of source performance than estimates based on shorter subsections of the plume.

**R:** Though in line 172, ERA-5 model resolution is now mentioned but this should be described in an earlier section (Section 2.4) too. I believe time interpolation is also made. Was the wind information applied to the exact emission position only and not for the downwind regions for the analysis? For this particular emission position how were the wind direction and speed interpolated in space and time from the original ERA-5 field?

**A:** A description of the ERA5 data products used was added to Section 2.4, including a description of the interpolation. "We used the ERA-5 hourly data on pressure levels (Hersbach et al., 2020), interpolated spatially using a 2D piecewise cubic approach from a 0.25°x 0.25°grid to each sites coordinates and temporally to the overpass of TROPOMI for each day".
Interpolation of these wind fields is a common step for this kind of analysis, see below:
https://doi.org/10.5194/essd-13-2995-2021
https://doi.org/10.5194/amt-13-2131-2020
https://doi.org/10.5194/acp-22-2745-2022
https://doi.org/10.5194/amt-13-205-2020

**R:** Lines 41-43. What is the relationship between the TROPOMI version 2.2.0 algorithm and the S5P-PAL product?

**A:** S5P-PAL takes the existing S5P dataset and reprocesses it using the v2.2.0 processor, so all the data used was produced using the same processor. A few clarifying remarks have been added to 41-45 to try to make this clearer.

**R:** Please check if your figures 1, 3 and 7 with maps/aerial images require a copyright statement/image credit and add it to the figures (or captions) (https://publications.copernicus.org/for_authors/manuscript_preparation.html#mapsaerials). If

these figures were entirely created by the authors, there is no need to add a copyright statement or credit. In that case it is important that you confirm this explicitly by email.

**A:** Figures 1 and 3 were produced by the authors using Python, with border information provided by NaturalEarth. Their website states: "No permission is needed to use Natural Earth. Crediting the authors is unnecessary." (https://www.naturalearthdata.com/about/terms-of-use/). Figure 7 uses map tiles provided by Stamen and OpenStreetMaps. We have added this credit to the figure caption. Please let us know if there is anything further required.